# Impaired Aversive Memory Formation in GPR37L1KO Mice

**DOI:** 10.3390/ijms232214290

**Published:** 2022-11-18

**Authors:** Vandana Veenit, Xiaoqun Zhang, Wojciech Paslawski, Ioannis Mantas, Per Svenningsson

**Affiliations:** Neuro Svenningsson, Department of Clinical Neuroscience, Karolinska Institutet, 171 76 Stockholm, Sweden

**Keywords:** GPR37L1KO, passive avoidance task, epinephrine, hippocampus, anxiety, GPR37L1 KD in the hippocampus, aversive related behavior

## Abstract

GPR37L1 is an orphan G-protein-coupled receptor, which is implicated in neurological disorders, but its normal physiological role is poorly understood. Its close homologue, GPR37, is implicated in Parkinson’s disease and affective disorders. In this study, we set out to characterize adult and middle-aged global GPR37L1 knock-out (KO) mice regarding emotional behaviors. Our results showed that GPR37L1KO animals, except adult GPR37L1KO males, exhibited impaired retention of aversive memory formation as assessed by the shorter retention latency in a passive avoidance task. Interestingly, the viral-mediated deletion of GPR37L1 in conditional knockout mice in the hippocampus of middle-aged mice also showed impaired retention in passive avoidance tasks, similar to what was observed in global GPR37L1KO mice, suggesting that hippocampal GPR37L1 is involved in aversive learning processes. We also observed that middle-aged GPR37L1KO male and female mice exhibited a higher body weight than their wild-type counterparts. Adult and middle-aged GPR37L1KO female mice exhibited a reduced level of serum corticosterone and middle-aged GPR37L1KO females showed a reduced level of epinephrine in the dorsal hippocampus in the aftermath of passive avoidance task, with no such effects observed in GPR37L1KO male mice, suggesting that lack of GPR37L1 influences behavior and biochemical readouts in age- and sex-specific manners.

## 1. Introduction

GPR37L1 is an orphan G-protein-coupled receptor enriched in oligodendrocyte progenitors and astrocytes [1,2,3]. Even though it is predominately expressed in the brain [4], GPR37L1 can be found in the heart and gastrointestinal tract [5,6,7]. GPR37L1 is known for its neuroprotective and glioprotective properties [8]. GPR37L1 shares more than 40 percent amino acid sequence homology with another G-protein-coupled receptor, GPR37, which is also expressed in the central nervous system [3,9,10,11]. It has been proposed that prosaposin exerts its neuroprotective actions through the activation of GPR37L1 and GPR37 [11]. Existing evidence in the literature suggests a link between GPR37 and Parkinson’s disease (PD), and GPR37 is proposed to aggregate in the cytosol of dopamine neurons from patients with PD [12,13]. Similarly, preclinical studies show that mice lacking GPR37 are associated with deficits in motor performance and synaptic plasticity, similar to what is observed in Parkinson’s patients [14,15]. GPR37 is also implicated in affective disorders [16,17], which is relevant in the context of PD, as many PD patients also exhibit depression and anxiety [18]. On the other hand, there is no evidence of GPR37L1 being involved in PD and other affective disorders. For example, one study shows that GPR37L1 knock-out (KO) mice do not exhibit any difference in motor-related behavior or exploratory behavior, as assessed by the open-field task and the novel object recognition task, respectively [19]. Other studies report that GPR37L1KO mice display advanced cerebellar development, accompanied by improved motor skills [19]. Moreover, GPR37L1 is linked with epilepsy [20]. Prosaptide-mediated GPR37L1 signaling exhibits neuroprotective actions in ischemic conditions by inhibiting astrocytic glutamate transporters and reducing N-methyl-D-aspartate receptor (NMDAR) activity [1]. Studies aiming to assess the physiological relevance of the GPR37L1 receptor in health and disease have revealed that GPR37L1 is associated with blood-pressure regulation, as GPR37L1KO mice show increased systolic blood pressure and hypertrophic heart compared to wild-type (WT) mice and mice that overexpress GPR37L1 in the heart [6].

The significant homology between GPR37 and GPR37L1, in combination with the possible involvement of GPR37 in affective behaviors [16,17], led us to investigate the role of GPR37L1 in emotional and cognitive behavior. Thus, we carried out a battery of behavior tests in GPR37L1KO mice (male and female), in both adult and middle-aged animals on the domains of anxiety, depression, and emotionally loaded memory processing. GPR37L1 is widely expressed in astrocytes [1]. The literature suggests that astrocytes are involved in ageing and in modulating fear-related memory function through adrenergic system [21,22]. We, therefore, also questioned whether astrocytic GPR37L1 in hippocampus is responsible for the observed behavioral phenotypes in middle-aged animals. To address this hypothesis, we knocked down (KD) GPR37L1 selectively in astrocytes located in dorsal hippocampus by viral-mediated deletion in 10–11-month-old conditional GPR37L1KO mice and carried out the passive avoidance task. Since there is evidence that astrocytic epinephrinergic/nor-epinephrinergic receptors modulate fear-related task performance [23], and the interaction between glucocorticoids and the norepinephrine system to modulate the retrieval of contextual fear memory [24], we assessed the levels of corticosterone in the serum and, epinephrine and nor-epinephrine levels in the dorsal hippocampus of these animals. Given the role of lactate released by hippocampal astrocytes in modulating emotionally loaded behaviors [25], we also examined lactate levels in the hippocampus of GPR37L1KO mice and compared them to their WT counterparts.

## 2. Results

### 2.1. Schematic of the Experimental Design

Figure 1 shows the experimental design, in which WT and global GPR37L1KO mice are subjected to a battery of behavior tests in both adulthood and middle-age before being sacrificed for biochemical analysis.

### 2.2. Effects on Locomotion, Anxiety-like and Depression-like Behavior in Adult and Middle-Aged WT and GPR37L1KO Male and Female Mice

To examine locomotion, anxiety-like, and depression-like behaviour, we carried out open-field test, elevated plus maze (EPM), light dark box and sucrose preference test. A three-way ANOVA on total distance covered in the open field test showed significant effects of age [F _(1,67)_ = 81.55, *p* < 0.0001; Figure 2A] and age x sex [F _(1,67)_ = 47.87, *p* < 0.0001; Figure 2A]. Post-hoc analysis using the two-stage linear step-up procedure of Benjamini, Krieger and Yekutieli showed adult WT and GPR37L1KO females travelled a greater distance compared to adult WT and GPR37L1KO male [(*p* = 0.005 and *p* = 0.005, respectively); Figure 2A]; middle-aged WT and GPR37L1KO males travelled a greater distance compared to middle-aged WT and GPR37L1KO females [(*p* = 0.003 and *p* < 0.0001, respectively); Figure 2A]; adult WT and GPR37L1KO females travelled a greater distance than middle aged WT and GPR37L1KO females [(*p* < 0.0001 and *p* < 0.0001, respectively); Figure 2A]. A three-way ANOVA on percent time of thigmotaxis behaviour revealed a significant effect of age [F _(1,67)_ = 9.50, *p* = 0.0030; Figure 2B], with no significant effect observed for other factors. Post-hoc analysis using the two-stage linear step-up procedure of Benjamini, Krieger and Yekutieli revealed that adult WT female mice spent more time in the corner than middle- aged WT female mice (Figure 2B, *p* = 0.0142) and adult GPR37L1KO males spent more time in the corner than middle-aged GPR37L1KO mice (Figure 2B, *p* = 0.045). A two-way ANOVA on percent time spent in the light chamber in light dark box test in adult WT and GPR37L1KO mice showed a significant effect of genotype [F _(1,33)_ = 6.68, *p* = 0.014], with no significant effect observed for sex or the interaction between genotype and sex. Post-hoc analysis using the two-stage linear step-up procedure of Benjamini, Krieger and Yekutieli showed that adult WT male showed a very strong tendency to spend more time in the light chamber of the light dark box compared to GPR37L1KO adult male (Figure 2C, *p* = 0.05). A two-way ANOVA on percent time spent in the open arms in the elevated plus maze test in middle-aged animals did not show any statistically significant difference in genotype, sex, or their interaction (Figure 2D, *p* = n.s.). A two-way ANOVA on the number of entries in the light chamber in the light dark box did not reveal any effect of genotype, sex or their interaction (Figure 2E, *p* = n.s.). A two-way ANOVA on the ratio of number of entries in the open arms to the total number of entries in the elevated plus maze (EPM) test revealed a significant effect of sex [F _(1,37)_ = 16.40, *p* = 0.0001); Figure 2F], with no significant difference observed for other factors (Figure 2F, *p* = n.s.). Post-hoc analysis using the two-stage linear step-up procedure of Benjamini, Krieger and Yekutieli showed that middle-aged WT females and middle-aged GPR37L1KO females exhibited a higher ratio of the number of entries in the open arms to the total number of entries compared to middle-aged WT males and GPR37L1KO males (*p* = 0.001 and *p* = 0.045, respectively; Figure 2F). To evaluate depression-like behaviour, we carried out a sucrose consumption test. A three-way ANOVA on the percent of sucrose that was consumed showed a significant effect of age x sex [F _(1,64)_ = 6.321, *p* = 0.014; Figure 2G]. Post-hoc analysis using the two-stage linear step-up procedure of Benjamini, Krieger and Yekutieli showed that adult WT males consumed more sucrose than middle-aged WT males (Figure 2G, *p* = 0.019); middle-aged WT males consumed less sucrose than middle-aged WT females [Figure 2G, *p* = 0.005].

### 2.3. Effect of GPR37L1 Deletion on Aversive Memory Processing

To examine aversive memory processing, animals were subjected to a passive avoidance task. A three-way ANOVA on the training latency did not reveal any significant effect of genotype, sex, age and its interactions. A three-way ANOVA on the retention latency revealed significant effect of age [F _(1,75)_ = 7.37, *p* = 0.008] and genotype [F _(1,75)_ = 14.91, *p* = 0.000], with no significant effect observed for other factors. Post-hoc analysis using the two-stage linear step-up procedure of Benjamini, Krieger and Yekutieli showed that adult GPR37L1KO females showed a lower retention latency compared to adult WT females (Figure 3, *p* = 0.015); middle-aged GPR37L1KO females showed a lower retention latency compared to middle-aged WT females (Figure 3, *p* = 0.035); middle-aged GPR37L1KO males showed a lower retention latency compared to middle-aged WT males (Figure 3, *p* = 0.010). Middle-aged GPR37L1KO male exhibited a lower retention compared to adult GPR37L1KO males (Figure 3, *p* = 0.024).

### 2.4. Effect of Astrocyte-Specific GPR37L1KD in the Hippocampus on Fear Related Behavior

Fluorescent in situ hybridization (ISH) experiments showed that GPR37L1 is enriched in Gfap-positive astrocytes in the dorsal hippocampus (Figure 4A). Conditional GPR37L1 (i.e., floxed mice) were injected with AAV-GFAP-Cre in dorsal hippocampus resulting in a 70% knockdown of the GPR37L1 in this region, detected with ISH (Figure 4B,C, t = 10.81, df = 25, *p* < 0.0001). Since both middle-aged male and female GPR37L1KO mice display a similar performance in passive avoidance task, we pooled the genders for the analysis of the knock down experiment (Figure 4D,E). An independent sample *t*-test did not show any statistical difference between the groups for the latency to enter the dark compartment on the training session of the passive avoidance task (*p* = n.s.). However, an independent sample *t*-test for the retention phase of the passive avoidance task revealed that astrocyte-specific GPR37L1KD animals exhibited a lower latency to enter the dark compartment (t = 2.130, df = 29, *p* = 0.041).

### 2.5. The Level of Epinephrine and Nor-Epinephrine in the Dorsal Hippocampus in Adult and Middle Aged WT and GPR37L1KO Male and Female Mice after PAT

A three-way ANOVA for the level of epinephrine in the dorsal hippocampus revealed a significant effect of sex [F _(1,46)_ = 12.60, *p* = 0.001; Figure 5A], and a trend towards significance for genotype [F _(1,46)_ = 2.94, *p* = 0.093; Figure 5A], with no significant effect observed for other factors (Figure 5A). Post-hoc analysis using the two-stage linear step-up procedure of Benjamini, Krieger and Yekutieli showed that middle-aged WT females exhibited a higher level of epinephrine in the dorsal hippocampus compared to middle-aged GPR37L1KO females (Figure 5A, *p* = 0.025); adult GPR37L1KO males exhibited higher epinephrine compared to adult GPR37L1KO females (Figure 5A, *p* = 0.002).

A three-way ANOVA for the level of norepinephrine in the dorsal hippocampus did not reveal any significant effect of age, sex, genotype, or their interaction (Figure 5B, *p* = n.s.).

### 2.6. Effect of GPR37L1KO on the Corticosterone Level after PAT

A three-way ANOVA on the level of circulating corticosterone revealed a significant effect of genotype x sex [F _(1,54)_ = 4.04, *p* = 0.049; Figure 6] and a strong trend towards significant effect of genotype [F _(1,54)_ = 3.66, *p* = 0.061; Figure 6], age x sex [F _(1,54)_ = 3.65, *p* = 0.061; Figure 6] with no significant effect observed for other factors (Figure 6). Post-hoc analysis using the two-stage linear step-up procedure of Benjamini, Krieger and Yekutieli showed that adult GPR37L1KO females showed a lower corticosterone compared to WT females (Figure 6, *p* = 0.047); middle-aged GPR37L1KO females showed lower corticosterone compared to middle-aged WT females (Figure 6, *p* = 0.038); middle-aged GPR37L1KO females showed lower corticosterone compared to middle-aged GPR37L1KO males (Figure 6, *p* = 0.022).

### 2.7. Effect of GPR37L1KO on Body Weight

A three-way ANOVA on the body weight revealed a significant effect of genotype [F _(1,100)_ = 19.08, *p* < 0.0001], age [F _(1,100)_ = 153.7, *p* < 0.0001], sex [F _(1,100)_ = 29.01, *p* < 0.0001] with no significant effect of other factors (Figure 7). Post-hoc analysis using the two-stage linear step-up procedure of Benjamini, Krieger and Yekutieli showed that adult WT males exhibited a higher body weight than adult WT females (Figure 7, *p* < 0.0001), adult GPR37L1KO males exhibited a higher body weight than adult GPR37L1KO females (Figure 7, *p* = 0.0003), and middle-aged WT males exhibited a higher body weight than middle-aged WT females (Figure 7, *p* = 0.0172). Interestingly, middle-aged GPR37L1KO male and female animals exhibited a higher body weight than middle-aged WT male and female animals, respectively (Figure 7, *p* = 0.034 and *p* = 0.004, respectively).

### 2.8. Effect of GPR37L1KO on the Level of Lactate in the Dorsal Hippocampus

Deletion of the GPR37L1 gene tended to decrease the level of lactate in the dorsal hippocampus (the samples were from both adult and middle-aged male and female mice for both the groups: WT and GPR37L1KO) (Figure 8). An independent sample *t*-test revealed that GPR37L1KO tends to have lower level of lactate in the dorsal hippocampus compared to their WT counterparts (t = 1.884, df = 29, *p* = 0.069).

## 3. Discussion

The current study provides evidence that GPR37L1 might be involved in mediating aversive memory. Adult GPR37L1KO female mice and middle-aged GPR37L1KO mice (both male and female) have impaired retention of aversive memory in the passive avoidance task. The region-selective knockdown of GPR37L1 in astrocytes of the hippocampus in middle-aged mice also leads to impaired retention in passive avoidance tasks. Adult GPR37L1KO female mice and middle-aged GPR37L1KO female mice exhibit significant reductions in the level of corticosterone level compared to their WT counterparts in the aftermath of the passive avoidance task. In addition to corticosterone, the level of epinephrine is also significantly lower in GPR37L1KO middle-aged female mice compared to their WT counterparts. No difference in the level of corticosterone is observed in GPR37L1KO male mice with respect to WT animals when examined after passive avoidance tasks. Our data also show that middle-aged GPR37L1KO mice have a higher body weight than their WT counterparts, and female mice have a lower body weight compared to male mice. There is an effect of GPR37L1 on some of the behavioral readouts in a sex- and age-specific manner.

Based on the fact that GPR37L1 is localized on astrocytes [1], and the role of astrocytes in fear-related behavior [23,26], we decided to investigate the behavior of GPR37L1KO mice on aversion and fear-related memory processing by carrying out passive avoidance tasks. Our passive avoidance task data highlight the role of GPR37L1 in memory formation for aversion in middle-aged animals and in adult female mice. This is an interesting observation, as astrocytes are involved in age-related memory impairment [22]. Interestingly, we obtained a similar finding in astrocyte-specific GPR37L1KD in middle-aged mice, suggesting that astrocyte-specific GPR37L1 might play a role in the memory retention of aversive stimuli. Given the role that the norepinephrine system in astrocytes of the dorsal hippocampus plays in modulating emotionally loaded behaviors [23], we examined the levels of epinephrine and norepinephrine in the dorsal hippocampus of GPR37L1KO male and female animals, along with WT animals in adulthood and in middle-age, by using high-performance liquid chromatography (HPLC). Our data show that middle-aged GPR37L1KO females have reduced levels of epinephrine compared to WT female animals in the dorsal hippocampus, with no corresponding difference observed in the level of norepinephrine. Both epinephrine and norepinephrine remained unchanged in GPR37L1KO males compared to WT in the dorsal hippocampus. Epinephrine acts on both beta-1(B1) and beta-2 (B2) epinephrinergic receptor, whereas norepinephrine acts only as a beta-1 epinephrinergic receptor. The fact that we observed reductions in the level of epinephrine (which acts through both B1 and B2 epinephrinergic receptors) and not nor-epinephrine (which acts only through B1 epinephrinergic receptor) in GPR37L1KO females fits well the existing findings in the literature, which show that astrocytic B2, and not B1 epinephrinergic receptor, KO animals also exhibit impaired retention memory for aversive stimuli [23]. Based on our finding, it is plausible to suggest that the GPR37L1 receptor might modulate astrocytic B2 epinephrinergic receptor in a sex-dependent manner. As well as epinephrine, we also observed reduced corticosterone levels in females following the passive avoidance task. There is evidence in the literature that the administration of corticosterone before training in passive avoidance tasks, or shortly after training, enhances long-term memory formation for passive avoidance tasks [27]. Moreover, the administration of corticosterone synthesis inhibitors such as metyrapone and aminoglutethimide shortly before training in the passive avoidance task inhibits the memory formation regarding the passive avoidance task [28]. Both epinephrine and corticosterone data in females fit well with the literature, which suggests that both epinephrine and cortisol treatment selectively enhance memory for emotionally arousing situations [29,30,31]. While corticosterone and epinephrine results fit well with the impaired retention of aversion-related tasks observed in GPR37L1KO females, we did not observe any significant difference in the level of epinephrine and the corticosterone in males. Hence, the impaired aversion memory retention observed in middle-aged male mice cannot be attributed to the level of corticosterone or epinephrine in GPR37L1KO male mice. It is possible that the deletion of GPR37L1 alters physiological and behavioral parameters in a gender-specific way. This view has been corroborated for cardiovascular parameters by a study showing that knocking out GPR37L1 alters cardiovascular parameters in a sex-specific way [10], suggesting that GPR37L1 contributes to sexual dimorphism of the central cardiovascular system [32].

Several studies also show a role of lactate released by astrocytes in the aftermath of passive avoidance task training in forming long-term memory [25]. As part of our pilot experiment, we also examined the level of lactate in dorsal hippocampus in these animals under basal conditions. We observed a tendency towards a decrease in the level of lactate in the dorsal hippocampus of GPR37L1KO animals compared to WT animals. These pilot data suggest that larger studies to examine sex differences in the role of GPR37L1 to regulate lactate levels are warranted.

The role of GPR37L1 gene has mostly been explored for motor-related behavior, with one study showing improved motor learning in mice lacking GPR37L1 [20] and the other study showing no difference in motor-related behavior in mice lacking GPR37L1 [1]. A study by [1] also shows no effect of the GPR37L1 gene on anxiety-related behavior, as assessed by open field tests and memory-related test and evaluated by novel object recognition. Our data on motor behavior in GPR37L1KO mice, as assessed by the open field test (Figure 2A), agree with [1], showing no effect of genotype on locomotion. However, there was a significant effect of age and age x sex on locomotion. While our data did not show any clear-cut effect of the GPR37L1 gene on the anxiety level, time spent in the light chamber in the light dark box does seem to suggest that GPR37L1 might play a role in modulating anxiety level in males. There was no effect of the GPR37L1 gene per se on the domain of depression.

The increased body weight observed in GPR37L1KO mice compared to their WT counterparts could possibly be due to GPR37L1’s association with blood pressure regulation, with GPR37L1KO mice showing an increase in systolic blood pressure and cardiac hypertrophy compared to mice with overexpressed GPR37L1 in the heart [22]. It is plausible that GPR37L1KO mice have a metabolic phenotype, which is often associated with increased body weight [33].

## 4. Concluding Remarks

Taken together, our results highlight the role of hippocampal GPR37L1 gene in mediating the memory formulation of aversion in aged animals and adult females. Our data suggest that the GPR37L1 gene influences behaviors in sex-specific and age-specific manners.

## 5. Materials and Methods

### 5.1. Subjects and Housing

The experiments were approved by the local ethical committee at Karolinska Institute (N139-16) and conducted in accordance with the European Communities Council Directive of 24 November 1986 (86/609/EEC). Mice were housed in temperature- and humidity-controlled rooms (20 °C, 53% humidity) in well-ventilated racks with a 12 h dark/light cycle. They had access to standard lab pellets and water ad libitum.

### 5.2. Breeding and Genotyping of Global GPR37L1 KO Mice

GPR37L1KO embryos on a C57BL/6J genetic background were a kind gift from Dr. Randy Hall. Animals were subsequently bred and maintained in the Wallenberg animal facility of Karolinska Institutet. First breeding was between GPR37L1KO male x C57BL6J female, and then heterozygote breeding was carried out until the experiments. Animals were genotyped to confirm genetic deletion of GPR37L1.

Genotyping was performed as previously described [34]. DNA for genotyping was extracted from tail or ear biopsy. The primers used are listed in Table 1. Briefly, DNA (2 µL) was amplified on a T100™ Thermal Cycler (#1861096, Bio-Rad Laboratories, Inc., Hercules, CA, USA) for 35 cycles (95 °C for 60 s, 58 °C for 30 s, and 72 °C for 60 s). After additional incubation at 72 °C for 10 min and being transferred to 4 °C, PCR products were subjected to electrophoresis in 1.5% agarose gel in GelRed. Relative intensity of PCR bands was analyzed using the ChemiDoc™ MP Imaging System (#12003154, Bio-Rad Laboratories, Inc., Hercules, CA, USA). The expected PCR fragment sizes are 366 bp for GPR37L1KO and 385 bp for WT. An example of agarose gel image is shown is Figure 9.

### 5.3. General Experimental Plan

To investigate the role of GPR37L1 on non-motor behavioral domains, a series of behavior studies were carried out on WT and GPR37L1KO male and female mice at two timepoints: adult (3–5 months) and middle-aged animals (11–14 months). The behavior tests for animals of different ages and sex were conducted at different timepoints. We did not assess the behavior in females at a particular estrous cycle, but we did ensure that the females were counterbalanced with respect to estrous cycles (assessed by vaginal smears prior to starting the series of behavior experiments) in WT and GPR37L1KO females to avoid any bias in the interpretation of results, as estrous cycle is known to influence the anxiety-like behavior of animals [35,36,37]. Most of the protocols of the behavior tests and sample analysis were followed as described in [38]. Behavior tests were carried out from the least to the most stressful to minimize influences from the previous test. Adequate resting days were given to the mice between tests. Behavior experiments were conducted between 9:00 and 14:00 h. The animals were euthanized by decapitation 30 min after the last behavior experiment, which was the passive avoidance test. Trunk blood was gathered for corticosterone analysis and the brains were rapidly fresh-frozen in isopentane and then stored at −80 °C. The brains were subsequently used for HPLC analysis by dissecting hippocampus (described below). The schematic of the experimental design is shown in Figure 1.

### 5.4. Behavioral Tests

#### 5.4.1. Open Field Test

The open field test is a common test for assessing locomotor activity and anxiety-like behavior in rodents [39,40]. Our primary goal of the open field test was to assess locomotion in GPR37L1KO mice; hence, the test was carried out for 15 min. The open-field arena was a square of 46 cm × 46 cm. The intensity of the light was maintained between 25 and 35 lux. Total distance travelled and time spent in the periphery was tracked and analyzed using an automated video tracking system (NOLDUS Ethovision XT11.5, Wageningen, The Netherlands). Thigmotaxis behavior in an open field arena was defined as the time spent in close vicinity of the walls.

#### 5.4.2. Elevated plus Maze

The protocol of elevated plus maze was followed as described in [41] to measure anxiety-like behavior. The elevated plus maze apparatus consisted of an elevated platform (50 cm above the ground) with two opposing open arms (25 × 5 cm), two opposing closed arms (25 × 5 × 15 cm) and a central platform (5 × 5 cm). The light intensity was set to 14–15 lux in the open arms, 5–10 lux in the center and 3–4 lux in the closed arms. Mice were placed individually in the maze facing one of the closed arms. Animals could explore the apparatus for 5 min. Time spent in each arm was measured and analyzed by an automated video tracking system (NOLDUS Ethovision XT11.5, The Netherlands).

#### 5.4.3. Light Dark Box

The light/dark exploration test was conducted to assess anxiety-like behavior [42]. The light dark box consisted of a dark compartment; 39 × 13 × 16 cm with a 13 × 8 cm aperture at floor level that opened onto a large white Plexiglas square arena (light compartment; 39 × 39 × 35 cm). In this test, mice were placed in a dark compartment of the light dark box. The light intensity in the light chamber was kept at around 120 lux. The number of entries in the light compartment (defined as all 4 paws out of the shelter) and time spent inside the light compartment over a 15 min session was calculated by an automated video tracking system (NOLDUS Ethovision XT11.5). Mice that did not enter the light compartment were assigned a 900 s latency to enter.

#### 5.4.4. Sucrose Preference Test

The sucrose preference or sucrose consumption test is a measure of anhedonia, based on the preference of a sucrose solution over water. Animals were individually housed in a cage and were presented with two bottles, one containing the 2% sucrose solution and one containing tap water for 3 days. The positions of the bottles were interchanged each day. The first 24 h of sucrose/water consumption was considered habituation and was, therefore, not considered when calculating the results. Sucrose preference was calculated as the percentage of the total volume consumed from 24 to 72 h.

#### 5.4.5. Passive Avoidance Task

The passive avoidance test is used to assess aversive memory processing [43]. The passive avoidance apparatus (25 cm × 50 cm × 25 cm) consisted of two equal sized compartments connected by a sliding door (7 cm × 7cm) (Ugo Basile, Comerio-Varese, Italy). The light intensity in the dark compartment was 2 lux, and 250 lux in the bright compartment. The passive avoidance test comprised training and testing. During training, the animal was placed in the bright compartment and could explore it for 60 s, after which the sliding door was opened, and the animal had a maximum of 300 s to enter the dark compartment. Once the mouse had entered the dark compartment, the sliding door was automatically closed and, after 3 s, a weak electrical stimulus (0.3 mA, 2 s scrambled current) was delivered through the grid floor. After 24 h, the animal was again gently placed in the light compartment, and the latency to enter the dark compartment was measured (retention latency) with a 540 s cutoff time. No electrical stimulus was given during the retention test. A significantly prolonged step-through latency during the test session indicates intact emotional contextual memory.

### 5.5. Corticosterone Assay

For determining serum corticosterone level, trunk blood was collected from the animals after decapitation in a regular Eppendorf tube. After 30–40 min of coagulation, the samples were centrifuged at 11,000 rpm for 10 min. Serum (supernatant) was transferred into another Eppendorf tube and stored at −80 °C until analysis. Corticosterone was measured using a commercially available ELISA kit (Enzo Life Sciences, Farmingdale, NY, USA), following the manufacturer’s instructions. Serum samples were diluted 1:10 in the assay buffer.

### 5.6. High Performance Liquid Chromatography (HPLC)

Animals were sacrificed under basal conditions or 30 min after passive avoidance task, and their brains were dissected, snap-frozen and stored at −80 °C. The level of neurotransmitters: epinephrine and norepinephrine in the hippocampus were measured by HPLC, as previously described [44].

Frozen brains were sectioned using a cryostat and 100 µm thick slices were mounted on slides and the region of interest, i.e., dorsal hippocampal (DH), was dissected using a scalpel blade. The tissue was collected in RNAse-free tubes.

Briefly, chemicals: L-Norepinephrine hydrochloride (NE), (±)-Epinephrine hydrochloride (EPI), acetonitrile (Chromasolv Plus), monobasic sodium phosphate, ethylene-diamine-tetra-acetic acid (EDTA) disodium salt, 1-octanesulfonic acid (OSA) sodium salt, triethylamine (TEA), 85% phosphoric acid, 70% perchloric acid (PCA), and sodium bisulfite were purchased from Sigma-Aldrich. HPLC-grade water was produced using a Milli-Q UltraPure water system. 

Hippocampal tissue was dissected, weighted, suspended in ice-caged 0.1 M PCA and immediately homogenized using an ultrasonic processor at 50% amplitude for 10 sec, until complete tissue disruption. Samples were incubated on ice for 10 min, vortexed, and centrifuged at 15,000× *g* for 10 min at 4 °C. The supernatants were transferred to spin columns with 0.2 µm nylon membrane filters and centrifuged at 10,000× *g* for 2 min. Obtained samples were immediately stored at −80 °C. All samples were injected into the HPLC within two weeks.

The analysis was performed on HPLC-ECD system (Dionex Ultimate 3000, ThermoFisher Scientific, Waltham, MA, USA). The separation was performed on a C18 reversed-phase column at 30 °C. The mobile phase (75 mM monobasic sodium phosphate, 1.7 mM OSA, 100 µL/L TEA, 25 µM EDTA, and 10% acetonitrile (*v/v*), pH 3.0), was pumped at a flow rate of 0.4 mL/min. The first and second analytical cells were set to −100 mV and +300 mV, respectively. Processed samples were thawed on ice about an hour before analysis, placed in the autosampler, and kept at 5 °C before injection. Chromatograms were acquired with Dionex Chromeleon 7 software. Analyte concentrations in tissue samples were expressed as ng/mg of frozen tissue.

### 5.7. GPR37L1KD in the Hippocampus

Conditional GPR37L1KO mice were made by introducing loxP sites flanking the first exon of the GPR37L1 gene (Cyagen, Santa Clara, CA, USA). Middle-aged mice (10–11 months aged, both male and female) conditional GPR37L1KO were anesthetized by a mix of ketamine/xylazine (100/10 mg/kg, i.p.) (Apoteket, Solna, Sweden), diluted in saline and were placed in a stereotaxic frame (Stoelting co., Wood Dale, IL, USA). Vector (AAV-control or AAV-Gfap-cre) (Addgene) solutions were injected bilaterally using a 5 μL Hamilton syringe. The coordinates for the injections were −2.1 mm (antero-posterior), −/+1.5 mm (medio-lateral) and −1.7 mm (dorso-ventral) relative to bregma (flat skull position). A total of 1 μL of viral vector solution (AAV-control or AAV-Gfap-cre) was infused at a rate of 0.2 μL per minute. The capillary was left in place for one minute, and slowly moved upwards. The capillary was cleaned between injections with 30% H_2_O_2_, 70% EtOH and dH_2_O.

Twelve weeks after the AAV injection, mice performed the passive avoidance test. The animals were sacrificed by decapitation; the brains were fresh-frozen in isopentane and then stored at −80 °C for further use.

### 5.8. In Situ Hybridization (ISH) Experiments

Fresh-frozen sections (12 μm thick) were prepared from the studies mice in a cryostat (Leica CM 3050 S) and used for in situ hybridization experiments, as previously described [45]. Briefly, 35S-labelled anti-sense and sense cRNA probes were prepared by in vitro transcription from cDNA clones corresponding to fragments of GPR37L1. The transcription was performed from 50–100 ng of linearized plasmid using [^35^S] UTP (1250 Ci/mmol) and T3 RNA polymerase. Sections were post-fixed in 4% PFA for 5 min at room temperature, rinsed twice in 4× sodium chloride/sodium citrate buffer (SSC) and placed into 0.25% acetic anhydride in 0.1 M triethanolamine/4× SSC (pH 8) for 10 min at room temperature. After dehydration in graded alcohols, the sections were hybridized overnight at 55 °C with 35S-labelled probe in 50 μL of hybridization solution (20 mM Tris—HCl/1 mM EDTA/300 mM NaCl/50% formamide/10% dextran sulphate/1× Denhardt’s/250 μg/mL yeast tRNA/100 μg/mL salmon sperm DNA/0.1% SDS/0.1% sodium thiosulphate). The slides were washed in 4× SSC (5 min, four times), RNAse A (20 μg/mL) (20 min, at 37 °C), 2× SSC (5 min, twice), 1× SSC (5 min), 0.5× SSC (5 min) at room temperature and rinsed in 0.1× SSC at 65 °C (30 min, twice) (all washes contained 1 mM DTT), before being dehydrated in graded alcohols. The slides were then exposed on X-ray films for from 4 to 28 days.

The Fluorescent ISH (RNAscope) was performed using the RNAscope Multiplex Fluorescent Assay (Advanced Cell Diagnostics), according to manufacturer’s protocol. In brief, cryostat (CM 3050 S, Leica, Deer Park, IL, USA) fresh-frozen thaw-mounted sections (12 μm thickness) were post-fixed in 4% PFA for 15 min at 4 °C and dehydrated in graded alcohols. Afterwards, Protease III (Advanced Cell Diagnostics, Newark, CA, USA) was applied for 30 min at 40 °C. Subsequently, the sections were hybridized with GPR37L1 probe (Mm-Gpr37l1, cat. 319301) for 2 h at 40 °C. After this step, four amplification steps were followed (Amp 1-FL 30 min at 40 °C, Amp 2-FL 15 min at 40 °C, Amp 3-FL 30 min at 40 °C, Amp 4B-FL 15 min at 40 °C). After the last amplification step, the sections were counterstained with DAPI (Advanced Cell Diagnostics) for 30 s and covered with coverslips using Dako fluorescent medium. Finally, the sections were imaged on an LSM 880 (Carl Zeiss, Jena, Germany) confocal microscope using a 20 × 0.8 NA or a 63 × 1.4 NA oil immersion objective. Z-stacks of 6–9 μm thickness were obtained in each caption.

### 5.9. Lactate Measurement in the Hippocampus

Lactate was measured using L-Lactate Assay Kit (Colorimetric) (ab65331) following an assay protocol provided in the kit. Animals under basal conditions from both studied ages were pooled for this analysis.

### 5.10. Statistical Analysis

Data were analyzed using three-way ANOVAs or two-way ANOVA or independent sample *t*-test as appropriate, using the statistical package GraphPad Prism 9 (GraphPad software Inc., San Diego, CA, USA). Data were checked for normality using the Shapiro–Wilk test. If the data failed in the normality test, they were log-transformed and subjected to the normality test again. Multiple comparisons were performed using the two-stage linear step-up procedure of Benjamini, Krieger and Yekutieli. The data are represented as mean ± SD. Significance was set at *p* < 0.05. Graphs were created using GraphPad Prism 9.

## Figures and Tables

**Figure 1 ijms-23-14290-f001:**
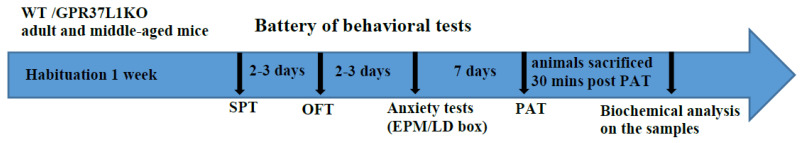
Schematics of the study show the timelines of the behavioral tests conducted in adulthood and in middle-aged WT and GPR37L1KO mice. SPT = Sucrose Preference Test, OFT = Open-Field Test, EPM = Elevated Plus Maze, LD box = Light Dark Box, PAT = Passive Avoidance Task.

**Figure 2 ijms-23-14290-f002:**
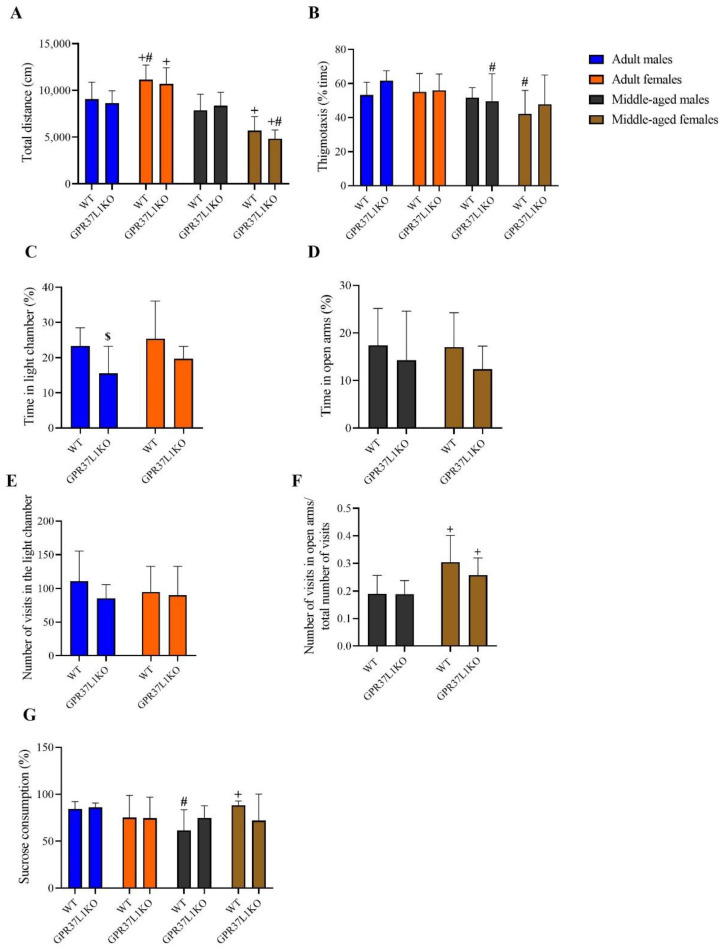
Effect on locomotion, anxiety and depression-like behaviour in adult and middle-aged WT and GPR37L1KO males and females. Total distance in open field (**A**). Percent time thigmotaxis behaviour in open field (**B**). Percent time spent in the light chamber in light dark box test (**C**). Percent time spent in the open arms in the elevated plus maze (**D**). Number of entries in the light chamber of the light dark box (**E**). Number of entries in the open arms of the elevated plus maze (**F**). Percent of sucrose consumption in sucrose preference test (**G**). *n*/group: 7 to 10. All data represent mean ± SD. ^#^ *p* < 0.05 versus corresponding age; ^+^ *p* < 0.05 with respect to sex; ^$^ *p* > 0.05 < 0.1 with respect to genotype calculated by three-way ANOVA or two-way ANOVA, followed by multiple comparisons using the two-stage linear step-up procedure of Benjamini, Krieger and Yekutieli.

**Figure 3 ijms-23-14290-f003:**
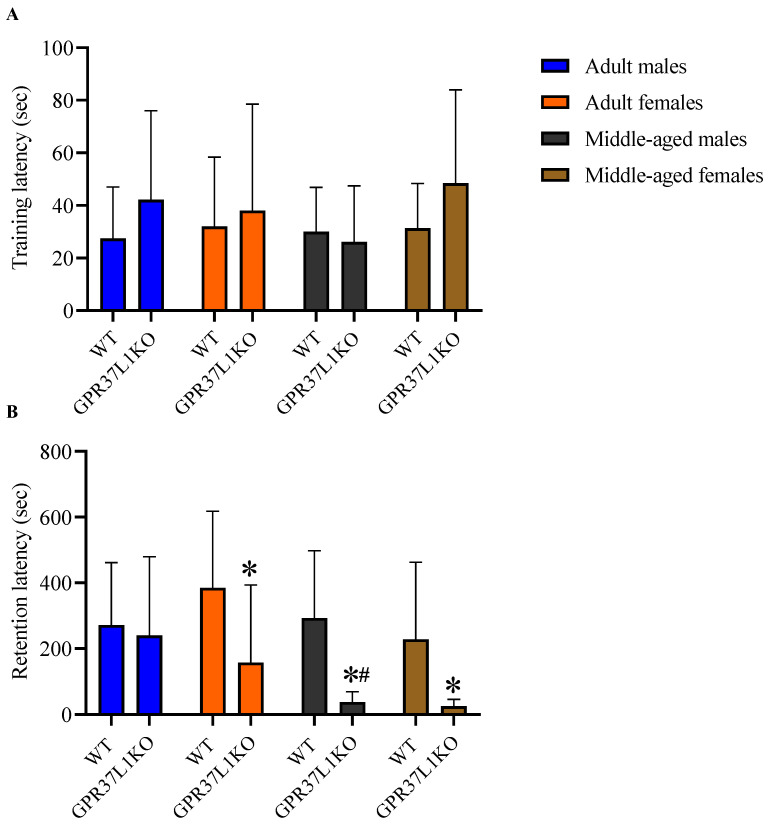
Effect on aversive memory in adult and middle-aged WT and GPR37L1KO males and females. Training latency (**A**) and retention latency (**B**). *n*/group: 7 to 12. All data represent mean ± SD. * *p* < 0.05 within genotype; ^#^ *p* < 0.05 versus corresponding age; calculated by three-way ANOVA followed by multiple comparisons using the two-stage linear step-up procedure of Benjamini, Krieger and Yekutieli.

**Figure 4 ijms-23-14290-f004:**
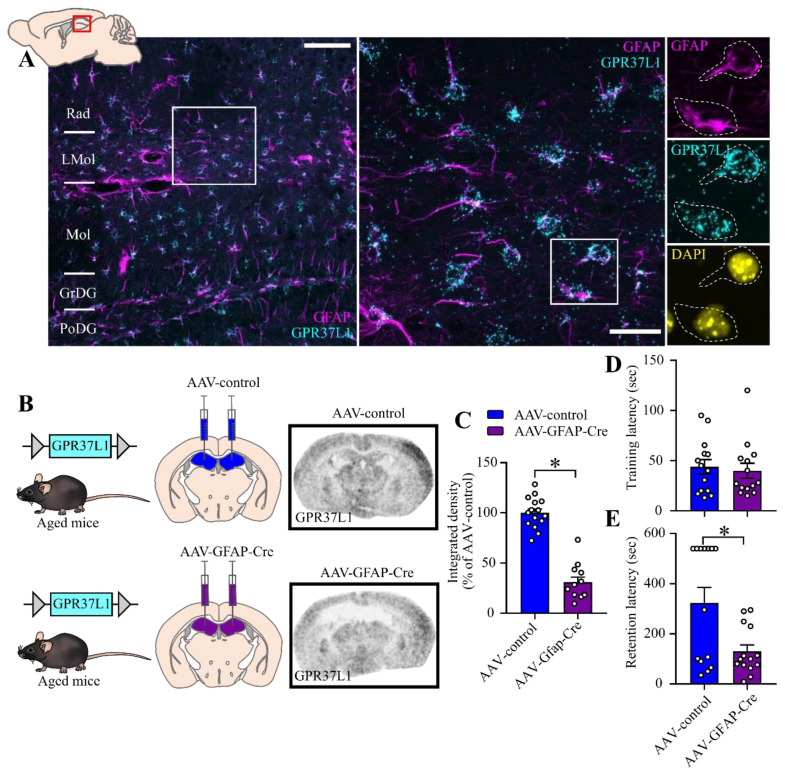
Effect of hippocampal GPR37L1 in the passive avoidance task (PAT) in hippocampal astrocyte-specific GPR37L1KD mice versus controls. Combined fluorescent ISH (RNAscope) and immunofluorescence showing the expression of GPR37L1 in GFAP-positive cells in dorsal hippocampus ((**A**), left scale bar: 100 µm, right scale bar: 30 µm). ISH confirmed the knockdown of the GPR37L1 gene in the hippocampus (**B**), quantification of GPR37L1 mRNA levels in the dorsal hippocampus (**C**), passive avoidance test for the mice that have been injected with AAV-control and AAV-Gfap-Cre, training phase of the passive avoidance task (**D**) and the retention phase of the passive avoidance task (**E**), *n* = 14–16/group. Statistical analysis was carried out using Student’s *t*-test. * *p* < 0.05. Results are expressed as Mean ± SEM. Rad: radial layer, LMol: lacunosum moleculare layer, Mol: molecular layer, GrDG: granular layer of dentate gyrus, PoDG: polymorph layer of dentate gyrus.

**Figure 5 ijms-23-14290-f005:**
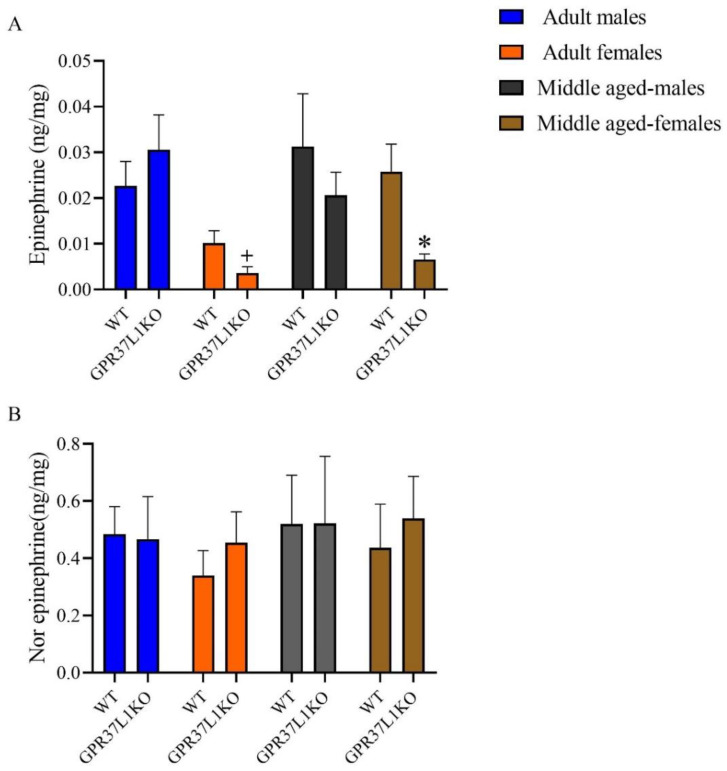
Effect on the level of epinephrine and nor epinephrine in the dorsal hippocampus in adult and middle-aged WT and GPR37L1KO males and females after PAT. Epinephrine in the dorsal hippocampus (**A**) Nor epinephrine in the dorsal hippocampus (**B**) *n*/group: 5 to 9. All data represent mean ± SD. * *p* < 0.05 within genotype; ^+^ *p* < 0.05 with respect to sex calculated by three-way ANOVA, followed by multiple comparisons using the two-stage linear step-up procedure of Benjamini, Krieger and Yekutieli.

**Figure 6 ijms-23-14290-f006:**
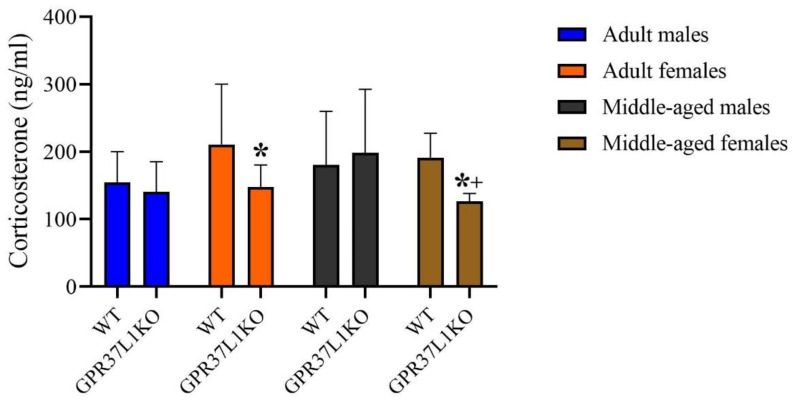
Effect on the level of circulating corticosterone in adult and middle-aged WT and GPR37L1KO males and females after PAT. *n*/group: 5 to 10. All data represent mean ± SD. * *p* < 0.05 within genotype; ^+^ *p* < 0.05 with respect to sex calculated by three-way ANOVA, followed by multiple comparisons using the two-stage linear step-up procedure of Benjamini, Krieger and Yekutieli.

**Figure 7 ijms-23-14290-f007:**
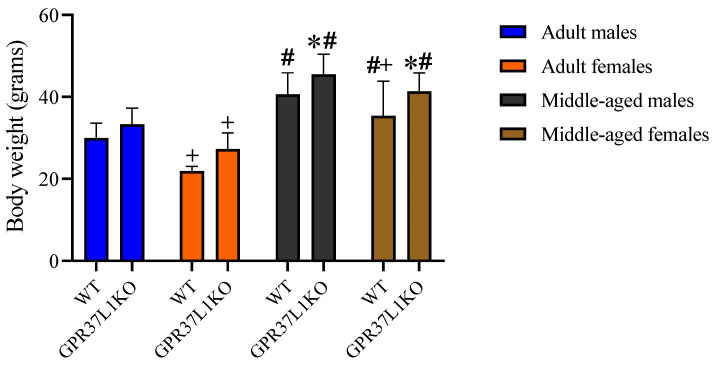
Effect on the level of body weight in adult and middle-aged WT and GPR37L1KO males and females. *n*/group: 8 to 12. All data represent mean ± SD. * *p* < 0.05 within genotype; ^#^ *p* < 0.05 versus corresponding age; ^+^ *p* < 0.05 with respect to sex calculated by three-way ANOVA, followed by multiple comparisons using the two-stage linear step-up procedure of Benjamini, Krieger and Yekutieli. WT = Wild Type, GPR37L1KO.

**Figure 8 ijms-23-14290-f008:**
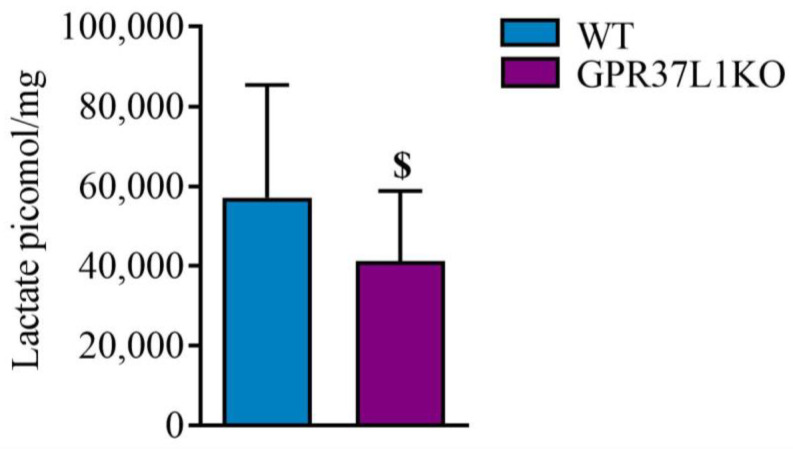
Effect of GPR37L1 deletion on the level of lactate in the dorsal hippocampus. *n* = 16/group. Statistical analysis was carried out using Student’s *t*-test. Results are expressed as Mean ± SD. ^$^ *p* > 0.05 < 0.1.

**Figure 9 ijms-23-14290-f009:**
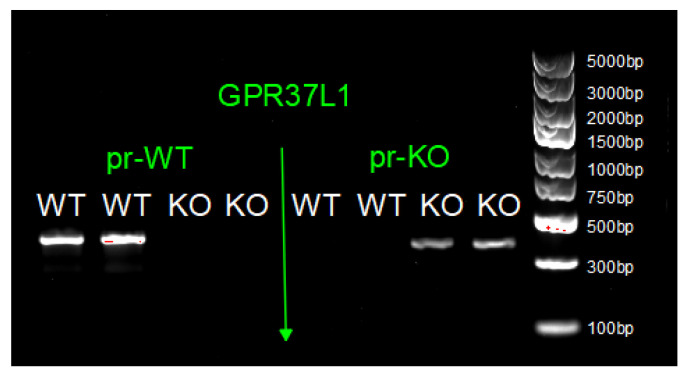
Agarose gel image shows the expected PCR fragments at 366 bp for GPR37L1KO and 385 bp for WT confirming the genotyping for the mice.

**Table 1 ijms-23-14290-t001:** Primers used for GPR37L1 genotyping.

Primer	Sequence, 5′–3′
GPR37L1KO, forward	GCAGCGCATCGCCTTCTATC
WT, forward	CACAGCTACTACTTGAAGAG
Common, reverse	ACACCTGCCTGTTCATCTGG

## Data Availability

All the data used in this study are available from the corresponding author upon request.

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
