# Peer review of "Impaired Aversive Memory Formation in GPR37L1KO Mice"

_ijms, 2022, doi:10.3390/ijms232214290_

Round 1
Reviewer 1 Report
The authors have investigated the role of GPR37L1 receptor in aversive memory formation and anxiety-, depressive-like behaviours in adult and middle aged male and female mice. To do so, they used a full KO of the GPR37L1 receptor. The results showed that middle aged mice showed impaired retention of memory formation in passive avoidance task. They demonstrated the involvement of the dorsal hippocampus by knocking down this gene in astrocytes. Norepinephrine was decreased in dorsal hippocampus in middle age KO GPR37L1 female mice and corticosterone in blood as well.
The purpose of the study is interesting and the design and methodology used are appropiate. However, the manuscript has big limitations such as lack of statistical accuracy which makes the significance of the data not entirely conclusive.
Major comments
- - Throughout the manuscript non-statistically significant results that land between 0.05 and 0.1 are represented as # in the figures and explained in the results as being “a trend towards significance”. This difficults the interpretation of the current findings and makes it hard to draw proper and real conclusions. Findings that result in p<0.1>0.05 cannot be used to draw any conclusion since they are not statistically significant. For example in the abstract, line 13-14 the authors conclude that all groups except adult male mice exhibit impaired retention of memory even if the adult females show only a trend.
- - The draft requires a revision on the statistical analysis. Data presented in the manuscript correspond to different genotypes, sex- and age- groups. Therefore, running a t-test is not indicated in this case.
- - Figure 3,figure 4. There is a lack of within subject comparison in the passive avoidance test. I suggest to make a latency ratio/index between training latency and retention latency. Then, evaluate if it follows normal distribution. If this is the case, the data should be statistically analyzed with a 2 way ANOVA ( age x genotype) or one way ANOVA accordingly.
- - A ISH quantitative measure demonstrating the knock down in the dorsal hippocampus should be provided. Based on the picture provided it doesn’t look like a knock down rather a knockout. Please also revise and clarify the concept “near-complete knock down” line 142.
- - Section 2.5 when results are explained group effect is mentioned. Clarify if it is refering to genotype.
- - 5B figure. Posthoc analysis is represented in the graph but is not explained in text.
- - Figure 6. # symbol not explained in the graph. 2 way ANOVA was not significant so post-hoc analysis cannot be run.
- - Figure 9. All groups should be represented and statistical analysis should be performed otherwise no conclusions can be made.
- - Discussion. Line 226-228. Contradiction:” Adult GPR37L1KO female mice and middle-aged GPR37L1KO mice (both male and female) have impaired retention of aversive memory processing in the passive avoidance task.
- - There is a lack of explanation on why/ how the lack of the GPR37L1 receptor might be affecting astrocytic B2 epinephrinergic receptor function in astrocytes
- - Methods section. Amygdala is mentioned in line 354 but not included in the study.
Reviewer 2 Report
A brief summary
In this manuscript, the behavioral characterization of GPR37L1KO mice and WT littermates was discussed in detail.
Broad comments
The main strength of the manuscript is the application of several behavioral methods. The main weaknesses are the flaws in the experimental design, methodology and statistical analysis.
Specific comments
Materials and Methods:
5.2. Breeding and Genotyping of global GPR37 KO mice
The expected PCR fragment sizes and the agarose gel images, proving the successful selection of KO mice, are missing.
Estrous cycle has a great impact on the anxiety-like behavior of female mice, according to several studies (Galeeva and Tuohimaa, 2001, https://doi.org/10.1016/S0166-4328(00)00341-7 , Walf et al., 2009, https://doi.org/10.1016/j.bbr.2008.09.016 , Gangitano et al., 2009, https://doi.org/10.1111/j.1601-183X.2009.00476.x ), etc. Which estrous (proestrus, estrus, metestrus or anestrus) phase did you select and why?
5.4.1. Open Field Test
Center/total ambulation distance % is missing.
Why did you choose 15 min for the open-field test? In the cited reference (40), 5 min was selected.
Open-field is an accepted test for assessing locomotor activity and anxiety-like behavior in rodents (see review by Prut, Belzung 2003, https://doi.org/10.1016/S0014-2999(03)01272-X). Both Materials and Methods and Results sections should be revised accordingly.
5.4.2. Elevated Plus Maze
Open arm entries (%) and total number of entries parameters are missing, from both Materials and Methods and Results sections.
5.10. Statistical analysis
Line 499: Which Student t-test you referred? Independent samples t-test, paired samples t-test or one sample t-test?
Results:
Data of male and female mice were evaluated separately. But data should be evaluated together to reveal the effect of sex. Same problem with the effect of age.
Moreover, in subsection 2.5., 2.6., the effects of genotype, treatment and their interaction were highlighted. I didn’t find any treatment in the manuscript.
In this experimental setup, there are three different factors (sex, age, genotype). The effects of sex, genotype, age and their interactions should be analyzed for subsections 2.2-2.6.
Throughout the Results section: Mean ± SEM is adequate way of presentation, only if the exact number of animals/group is also displayed. “N = 5-10/group” is not acceptable, when you use Mean ± SEM.
Discussion
Line 298-308: this paragraph should be revised after the re-evaluation of the behavioral data.
Round 2
Reviewer 1 Report
The changes introduced by the authors have significantly improved the manuscript but there are few things that still need to be addressed before publication.
1) One of the most revealing and interesting results of the manuscript is the experiment shown in figure 4, where the authors demonstrate that the specific KD of GPR37L1 in the astrocytes of the dorsal hippocampus significantly decrease the retention latency in the passive avoidance test.
It is not stated anywhere in the result section that the knockdown is specific to astrocytes. It should be clearly stated that is specific to astrocytes in the results headings, main text of results, figure title and figure text.
2) Figure 8 shows the level of lactate from the dorsal hippocampus. Throughout the paper the authors have investigated the sex and genotype effect on their interventions. Please specify why the authors pool the data.
Reviewer 2 Report
5.4.2. Elevated Plus Maze
Total number of entries parameter (also called total activity, defined as the total number of crosses between any arms) is still missing, from both Materials and Methods and Results sections. This parameter is also essential for the interpretation of the EPM results.
Results:
Throughout the Results section: the exact p value should be displayed for every comparison, displaying p < 0.05 is not informative.
Degrees of freedom should be displayed in sub script, for example F(1,35) = 15.17.
